# Unified theory for light-induced halide segregation in mixed halide perovskites

Zehua Chen [1,2], Geert Brocks[1,2,3], Shuxia Tao [1,2✉] & Peter A. Bobbert [2,4✉]

Mixed halide perovskites that are thermodynamically stable in the dark demix under illumination. This is problematic for their application in solar cells. We present a unified thermodynamic theory for this light-induced halide segregation that is based on a free energy lowering of photocarriers funnelling to a nucleated phase with different halide composition and lower band gap than the parent phase. We apply the theory to a sequence of mixed iodine-bromine perovskites. The spinodals separating metastable and unstable regions in the composition-temperature phase diagrams only slightly change under illumination, while light-induced binodals separating stable and metastable regions appear signalling the nucleation of a low-band gap iodine-rich phase. We find that the threshold photocarrier density for halide segregation is governed by the band gap difference of the parent and iodine-rich phase. Partial replacement of organic cations by cesium reduces this difference and therefore has a stabilizing effect.

[1] Materials Simulation and Modelling, Department of Applied Physics, Eindhoven University of Technology, Eindhoven, The Netherlands. [2] Center for Computational Energy Research, Department of Applied Physics, Eindhoven University of Technology, Eindhoven, The Netherlands. [3] Computational Materials Science, Faculty of Science and Technology and MESA+ Institute for Nanotechnology, University of Twente, Enschede, The Netherlands. [4] Molecular Materials and Nanosystems, Eindhoven University of Technology, Eindhoven, The Netherlands. ✉email: S.X.Tao@tue.nl; P.A.Bobbert@tue.nl

Metal-halide perovskites have extraordinary optoelectronic performance in solar energy harvesting and light emission applications[1–8]. The flexibility of the perovskite ABX$_3$ structure—where A is an organic or inorganic cation, B is a metal cation like Pb or Sn, and X is a halide anion like I or Br—allows to stabilize a preferred phase and tune the band gap through compositional alloying on different lattice sites[9–19]. Notably, mixed halide perovskites have been successfully used in tandem solar cells, where the band gaps in the two light-absorbing layers should be optimally tuned[18,20,21]. However, one of the biggest problems in mixed halide perovksites is their photoinstability, specifically light-induced halide segregation[17,22–24]. Studies on mixed halide perovskites show that exposure to continuous illumination leads to the separation of the different halide ions, resulting in the formation of low- and high-band gap domains. The low-band gap domains act as photocarrier traps, as evidenced by redshifts in photoluminescence observations[17,22–24]. The demixing is reversible because when kept in the dark for a sufficient amount of time, the perovskites return to their original mixed state. An important finding is the existence of a threshold intensity for halide segregation[23,25]. The reversibility and the existence of an illumination threshold suggest a thermodynamic origin of the effect.

Many strategies have been proposed to suppress this halide segregation, such as enhancing grain size and improving overall crystallinity[26], reducing carrier diffusion lengths[23], partial substitution of Pb with Sn[27], application of external pressure[28], alloying Cl into the I/Br lattice[29], and A-cation alloying[6,30–32]. A-cation alloying has attracted significant attention and has proven to be effective against halide segregation. It has been found that MA/Cs, FA/Cs, and FA/MA/Cs mixed halide perovskites (MA stands for methylammonium and FA for formamidinium) exhibit a reduced tendency for halide segregation[6,28,30–34]. In ref. [33] the reduced tendency for halide segregation when replacing MA by Cs was attributed to the smaller polarizability of Cs$^+$ as compared to MA$^+$, which would reduce electron-phonon coupling and suppress halide segregation. In a previous study, we have shown that A-cation alloying can change the band gap by changing the volume of the ABX$_3$ unit cell or by introducing octahedral distortions[35]. These structural deformations change the hybridization between the B- and X-site ions, which changes the conduction band minimum and valence band maximum energies. These electronic changes resulting from A-cation alloying may also contribute to phase stability improvements.

Several explanations have been given for light-induced halide segregation. Polaron-induced strain gradients under illumination have been suggested to drive the nucleation of low-band gap iodine-rich domains[36,37]. Other studies suggest that local electric fields caused by electron-hole pairs in the thin film[38] or at the surface[39] are the driving force for ion migration and demixing. It has also been proposed that a strong gradient in carrier generation rate through the film thickness can be a driving force for halide segregation[40]. These explanations, however, do not account for the observed existence of a threshold illumination intensity for halide segregation. A model based on band gap differences between perovskites with different halide compositions does account for a threshold illumination intensity[23]. In that model, it is suggested that the band gap difference between mixed I/Br and I-rich domains, where photocarriers can reduce their free energy by funneling to the I-rich domains, is the driving force behind the demixing. When applied to MAPb(I$_{0.5}$Br$_{0.5}$)$_3$, the model yields an illumination threshold that is of the same order of magnitude as found in a recent experiment, but it leaves the observed strong temperature dependence of the threshold unexplained[25].

There is a clear need for a unified theory for light-induced halide segregation in mixed halide perovskites that is transferable

and flexible. Understanding is lacking about the influence of temperature on the illumination intensity threshold for halide segregation, but also about the role of material composition. For example, we are not aware of explanations for the improved photostability after partial Cs substitution in MAPb(I$_{1-x}$Br$_x$)$_3$ and FAPb(I$_{1-x}$Br$_x$)$_3$[6,28,30–34]. From this, we conclude that it is not understood how A-site alloying influences the thermodynamic stability under illumination. Such understanding is needed to avoid or mitigate the effect.

Here, we provide such a unified theory and apply it to light-induced halide segregation in MAPb(I$_{1-x}$Br$_x$)$_3$, FAPb(I$_{1-x}$Br$_x$)$_3$, CsPb(I$_{1-x}$Br$_x$)$_3$, MA$_{7/8}$Cs$_{1/8}$Pb(I$_{1-x}$Br$_x$)$_3$, and FA$_{7/8}$Cs$_{1/8}$Pb(I$_{1-x}$Br$_x$)$_3$, which are experimentally among the most studied perovskite compounds. We have added the latter two compounds to our study to investigate the influence of partial Cs substitution. The free energy for each compound in the dark is determined using binary alloying theory. By adding a contribution to the free energy from photocarriers, we obtain a parameter-free theory that enables us to construct the phase diagrams for each compound in the dark and under illumination, distinguishing stable, metastable, and unstable regions. The theory also enables us to determine the dependence of the illumination threshold for halide segregation on temperature and material composition.

## Results

**Free energy in the dark.** We first consider the compositional Helmholtz free energy of the five compounds, applicable to the situation in the dark, within the quasi-chemical approximation (QCA, see 'Methods') of binary alloying theory[41]. This starts with a calculation of the configurational mixing enthalpy $\Delta U$ of the compounds (see Eq. (5) in 'Methods'). For this, we calculate within density functional theory (DFT) the energy of possible configurations of the halide anions in a supercell geometry. For the single-cation compounds MAPb(I$_{1-x}$Br$_x$)$_3$, FAPb(I$_{1-x}$Br$_x$)$_3$, and CsPb(I$_{1-x}$Br$_x$)$_3$ we take supercells with two formula units, following Brivio et al.[42], while for the double-cation compounds MA$_{7/8}$Cs$_{1/8}$Pb(I$_{1-x}$Br$_x$)$_3$ and FA$_{7/8}$Cs$_{1/8}$Pb(I$_{1-x}$Br$_x$)$_3$ we take supercells with eight formula units. Figure. 1a–e display for each of the five compounds the configurations with the lowest mixing enthalpy for $x = 0.5$. The mixing enthalpies per formula unit (f.u.) for the possible configurations of the I and Br anions in the supercell are given by circles in Fig. 1f–j.

By applying the QCA we obtain from the mixing enthalpies of the possible configurations at discrete relative Br concentrations $x = 0, 1/6, 1/3, 1/2, 2/3, 5/6, 1$ the mixing enthalpy $\Delta U(x, T)$ as a continuous function of $x$. The curves in Fig. 1f–j show $\Delta U(x, T)$ for different temperatures $T$ in the range 150–350 K. Figure 1k–o display the mixing free energy $\Delta F(x, T) = \Delta U(x, T) - T\Delta S(x, T)$ per formula unit, where $\Delta S(x, T)$ is the configurational mixing entropy.

In the single-cation materials, the width of the distribution in the mixing enthalpy $\Delta U$ for the different halide configurations increases in the order Cs–MA–FA of increasing cation size. The explanation is that, because of the different sizes of the halide anions (I is bigger than Br), the strain in the lattice for the different halide configurations is best accommodated for by CsPb(I$_{1-x}$Br$_x$)$_3$, followed by MAPb(I$_{1-x}$Br$_x$)$_3$ and FAPb(I$_{1-x}$Br$_x$)$_3$. The increasing cation size in this sequence is also reflected in the order of increasing unit cell volumes of the three compounds (see Supplementary Note 1). The incorporation of Cs in a relative concentration of 1/8 in MA$_{7/8}$Cs$_{1/8}$Pb(I$_{1-x}$Br$_x$)$_3$ and FA$_{7/8}$Cs$_{1/8}$Pb(I$_{1-x}$Br$_x$)$_3$ shows an expected slight decrease of the width of the distribution as compared to MAPb(I$_{1-x}$Br$_x$)$_3$ and FAPb(I$_{1-x}$Br$_x$)$_3$. The width of the distribution in the mixing enthalpy affects the symmetry $x \to 1 - x$ in the mixing free energies curves,

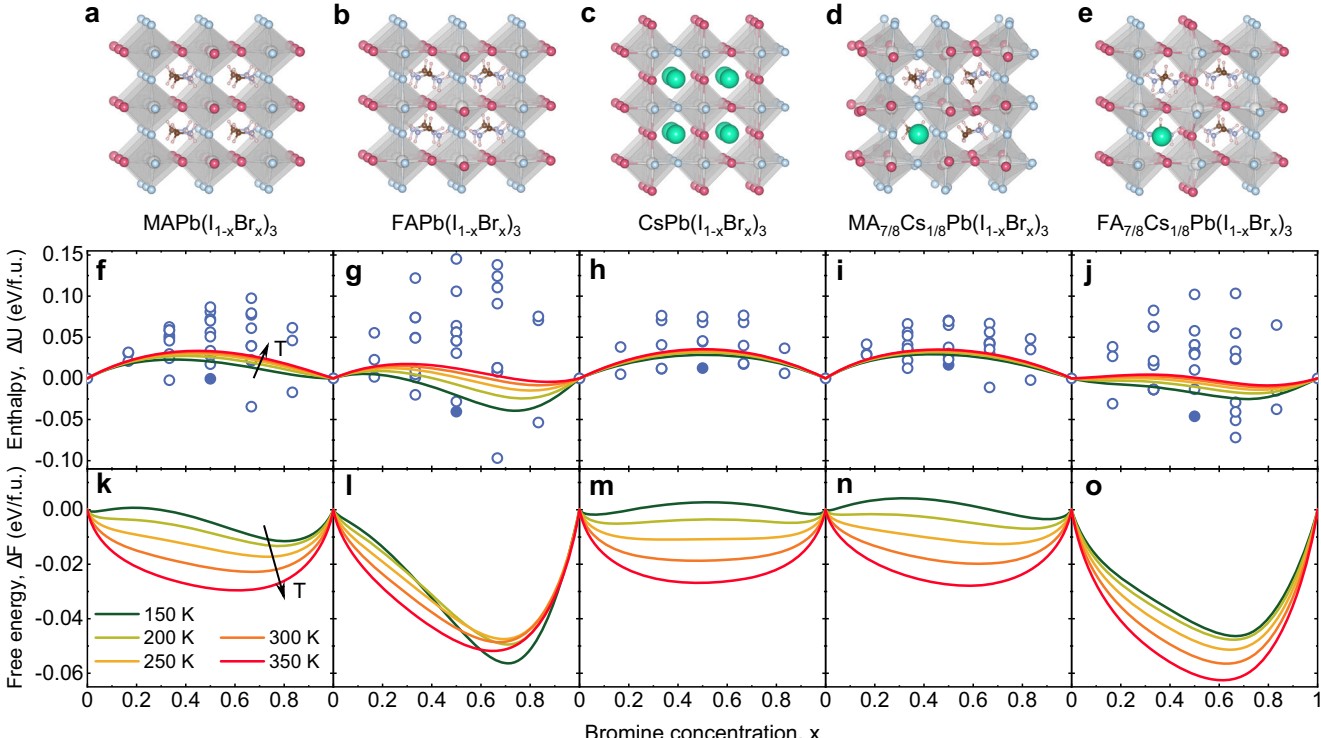

**Fig. 1 Mixing enthalpy and free energy of mixed I/Br perovskites in the dark. a–e** Atomic structures of the most stable configurations of the different compounds at relative Br concentration $x = 0.5$. Red spheres: I. Blue spheres: Br. White spheres inside octahedra: Pb. Green spheres: Cs. Cationic molecules in between octahedra: methylammonium (MA) or formamidinium (FA). **f–j** Mixing enthalpy per formula unit (f.u.) as a function of Br concentration. Circles: values calculated for each mixed configuration. Filled circles: values for the most stable configurations at $x = 0.5$, displayed in (**a–e**). Curves: results for the quasi-chemical approximation (QCA) at different temperatures. **k–o** Mixing free energy per formula unit as a function of Br concentration.

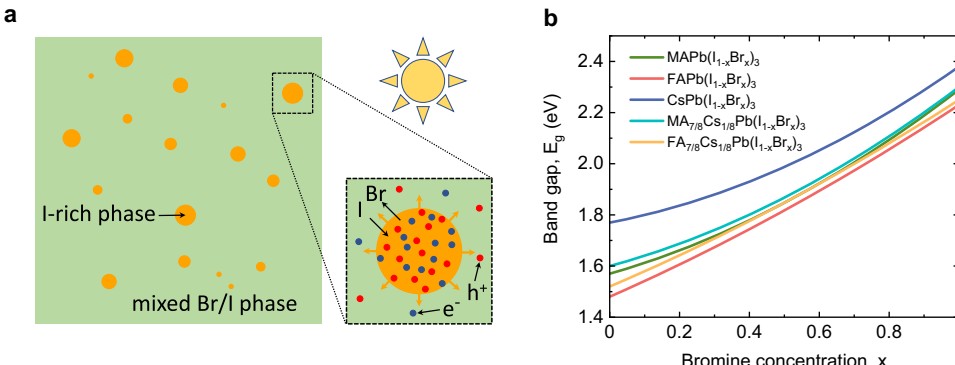

**Fig. 2 Mechanism of light-induced halide segregation. a** Nucleation of an I-rich phase from a mixed I/Br phase under illumination. The compositional free energy favours mixing, but this is dominated by a free energy decrease favouring demixing due to funneling of electrons and holes into low-band gap I-rich nuclei that grow by inward diffusion of I and outward diffusion of Br. **b** Band gap as a function of relative Br concentration $x$ for the different compounds.

with the curves for $CsPb(I_{1-x}Br_x)_3$ being the most and those for $FAPb(I_{1-x}Br_x)_3$ the least symmetric. This has an effect on the symmetry of the phase diagrams, as we will show further on.

**Light-induced halide segregation.** The key ingredient of our unified theory for light-induced halide segregation is the consideration of the combination of the compositional free energy in the dark and the free energy of photocarriers in the presence of illumination. Photocarriers can reduce their free energy by funneling to low-band gap domains[23], which leads to a driving force for halide demixing, as illustrated in Fig. 2a. It is well known that halide anions in metal-halide perovskites are quite mobile.

Because of stochastic fluctuations in halide composition, I-rich regions with lower band gap than the parent phase will spontaneously arise. Accumulation of diffusing photocarriers in these regions will reduce the photocarrier free energy. The free energy can then be further reduced by the growth of these regions by inward diffusion of I, leading to nucleation of an I-rich phase and to phase separation. Because the stoichiometry cannot change, the inward diffusion of I should be accompanied by the outward diffusion of Br.

In our theory, band gap differences of perovskites with different halide compositions play a central role. Figure 2b shows the band gaps as a function of Br concentration $x$ for the five compounds. The band gaps of the three single-cation perovskites

are obtained from experiment[17–19], while the band gaps of the two double-cation perovskites are obtained from an interpolation scheme (see Supplementary Note 2). The differences in band gap for different halide compositions are mainly caused by differences in the energy of the valence band maxima, where an increase in Br concentration decreases the energy of the valence band maximum[23]. So, it will be mainly the photogenerated holes that can reduce their free energy by funneling into I-rich domains. The electrons will follow the holes to establish local charge neutrality. The steepness of the band gap curves decreases for different cation compositions in the order FA–FA$_{7/8}$Cs$_{1/8}$–MA–MA$_{7/8}$Cs$_{1/8}$–Cs, following the decreasing trend in the (average) cation size. The decrease in the steepness of the band gap curves with Cs alloying may look surprising because the A-site cation does not directly contribute to the states governing the band gap. However, A-cation alloying can change the band gap indirectly by changing the volume of the unit cell or by introducing octahedral distortions[35].

The total system consisting of the perovskite with its distribution of halide anions and the generated photocarriers will try to lower its free energy by combined motion of halide anions and photocarriers. The free energy of the total system is the compositional free energy of the perovskite for a certain distribution of the halide anions (the free energy in the dark) and the free energy of a certain distribution of the photocarriers. Because the diffusion of photocarriers is much faster than that of the halide anions, the distribution of the carriers over the different phases will be at any moment in time in equilibrium. We define $n$ as the density of photogenerated electrons or holes per formula unit in the mixed phase. When $n \ll 1$, we can use Boltzmann instead of Fermi-Dirac statistics for the distribution of photocarriers. If phase separation occurs into two phases with Br concentrations $x_1$ and $x_2$, the photocarriers will redistribute over these two phases according to the Boltzmann factors $\exp\left(-E_g(x_1)/k_B T\right)$ and $\exp\left(-E_g(x_2)/k_B T\right)$, where $E_g(x)$ is the band gap as a function of Br concentration $x$ and $k_B T$ the thermal energy. This leads to

$$\frac{n_2}{n_1} = \exp\left(-\left(E_g(x_2) - E_g(x_1)\right)/k_B T\right), \tag{1}$$

where $n_1$ and $n_2$ are the photocarrier densities in the two phases. With $\phi_1$ and $\phi_2$ the corresponding volume fractions of the two phases, the mixing free energy $\Delta F^\star$ per formula unit under illumination then becomes

$$\Delta F^\star(x_1, x_2, \phi_1, \phi_2, T) = \phi_1 \Delta F(x_1, T) + \phi_2 \Delta F(x_2, T) + n_1 \phi_1 E_g(x_1) + n_2 \phi_2 E_g(x_2). \tag{2}$$

Neglecting the small volume difference per formula unit between the two phases, the conditions $\phi_1 + \phi_2 = 1$ and $\phi_1 x_1 + \phi_2 x_2 = x$ should hold. The sum of the first and second terms in Eq. (2) is the volume-weighted compositional mixing free energy in the dark. The sum of the third and fourth terms is the photocarrier contribution to the free energy. Because the band gap difference between the I-rich and parent phase is in general much larger than the thermal energy ($k_B T \approx 25$ meV at room temperature), even a low illumination intensity can according to Eq. (1) lead to a relatively large change of $\Delta F^\star$, which manifests the funneling effect.

In steady state, the rate of generation of photocarriers in the system should be equal to the sum of the rates of photocarrier annihilation by monomolecular and bimolecular recombination in the different phases:

$$G = \phi_1 \left(n_1/\tau + k n_1^2/V\right) + \phi_2 \left(n_2/\tau + k n_2^2/V\right). \tag{3}$$

Here, $G$ is the photocarrier generation rate per formula unit, which is proportional to the illumination intensity. The monomolecular and bimolecular recombination rates are given by an inverse photocarrier lifetime $\tau$ and a bimolecular recombination rate constant $k$, for which we take $\tau = 100$ ns and $k = 10^{-10}$ cm$^3$ s$^{-1}$, applicable for a standard MAPbI$_3$ film[43]. For the volume per formula unit $V$ we take the value $2.5 \times 10^{-22}$ cm$^3$ for MAPbI$_3$ (see Supplementary Note 1).

Equations (1–3) are the basis of our unified theory for light-induced halide segregation. With Eqs. (1) and (3), $n_1$ and $n_2$ can be calculated for a given $G$. Insertion in Eq. (2) then yields the mixing free energy under illumination, from which the spinodal and binodal for halide phase segregation can be obtained (see 'Methods'). We note that Eqs. (1–3) are only generally applicable if the diffusion lengths of the photogenerated holes and electrons are large compared to the feature sizes of the demixing. In that case, an equilibrium distribution of holes and electrons is established over the two phases throughout the system, with a homogeneous photocarrier density in each phase. However, we will use these equations only to determine the onset of phase separation. The sizes of the nuclei are then much smaller than the diffusion lengths so that the distribution of holes and electrons within the nuclei is homogeneous. The distribution in the parent phase can then still be inhomogeneous, but this does not affect the applicability of our theory to the onset of phase separation (see Supplementary Note 3).

**Phase diagrams.** In Fig. 3a–e we show the $x$-$T$ phase diagrams for the five mixed compounds in the dark, obtained from Eq. (2) for vanishing photocarrier density. The red lines are the spinodals separating the metastable (grey) and unstable (pink) regions. The blue lines are the binodals separating the stable (white) and metastable regions. Apart from FA$_{7/8}$Cs$_{1/8}$Pb(I$_{1-x}$Br$_x$)$_3$, miscibility gaps appear below the critical points ($x_c$, $T_c$), where the critical temperature $T_c$ is below room temperature. This means that at room temperature the mixed compounds are thermodynamically stable. The critical temperatures decrease for different cation compositions in the order MA–FA–Cs–MA$_{7/8}$Cs$_{1/8}$. The results show that changing MA by FA as well as mixing in Cs in the MA and FA compounds has a stabilizing effect in the dark. The amount of asymmetry in the phase diagrams under the change $x \to 1 - x$ in Fig. 3a–e is in accordance with the amount of asymmetry in the free energy curves shown in Fig. 1k–o. The compound FA$_{7/8}$Cs$_{1/8}$Pb(I$_{1-x}$Br$_x$)$_3$ is special in the sense that it is stable in the dark for all values of $x$ and $T$. Like MA$_{7/8}$Cs$_{1/8}$Pb(I$_{1-x}$Br$_x$)$_3$, the free energy curves are strongly asymmetric, but in contrast to MA$_{7/8}$Cs$_{1/8}$Pb(I$_{1-x}$Br$_x$)$_3$ no points of common tangent or inflection points occur, which would be the locations of the binodals and spinodals, respectively; see Fig. 1l and o. We checked that this situation does not change for $T < 150$ K, which is the lowest temperature in Fig. 1, so that there is no critical temperature.

We note that in the case of MAPb(I$_{1-x}$Br$_x$)$_3$ our phase diagram in the dark differs from that of Brivio et al.[42]. In particular, our $T_c$ of 266 K is below the value of 343 K in ref. [42]. The reason for the difference is that in ref. [42] the symmetry lowering by the specific orientation of the MA cations is neglected, leading to a reduction of the number of considered different configurations of the halide anions. The finding that our $T_c$ is below room temperature is in agreement with the observation that MAPb(I$_{1-x}$Br$_x$)$_3$ does not phase separate at room temperature in the dark[22]. This comparison does show that subtle differences in the way the free energy is calculated can have a substantial influence on the phase diagram. Therefore, also our phase diagrams can have inaccuracies that are related to, e.g. the specific exchange-correlation functional used in the DFT calculations (see 'Methods') and the limited size of the used supercells. Also, thermally induced

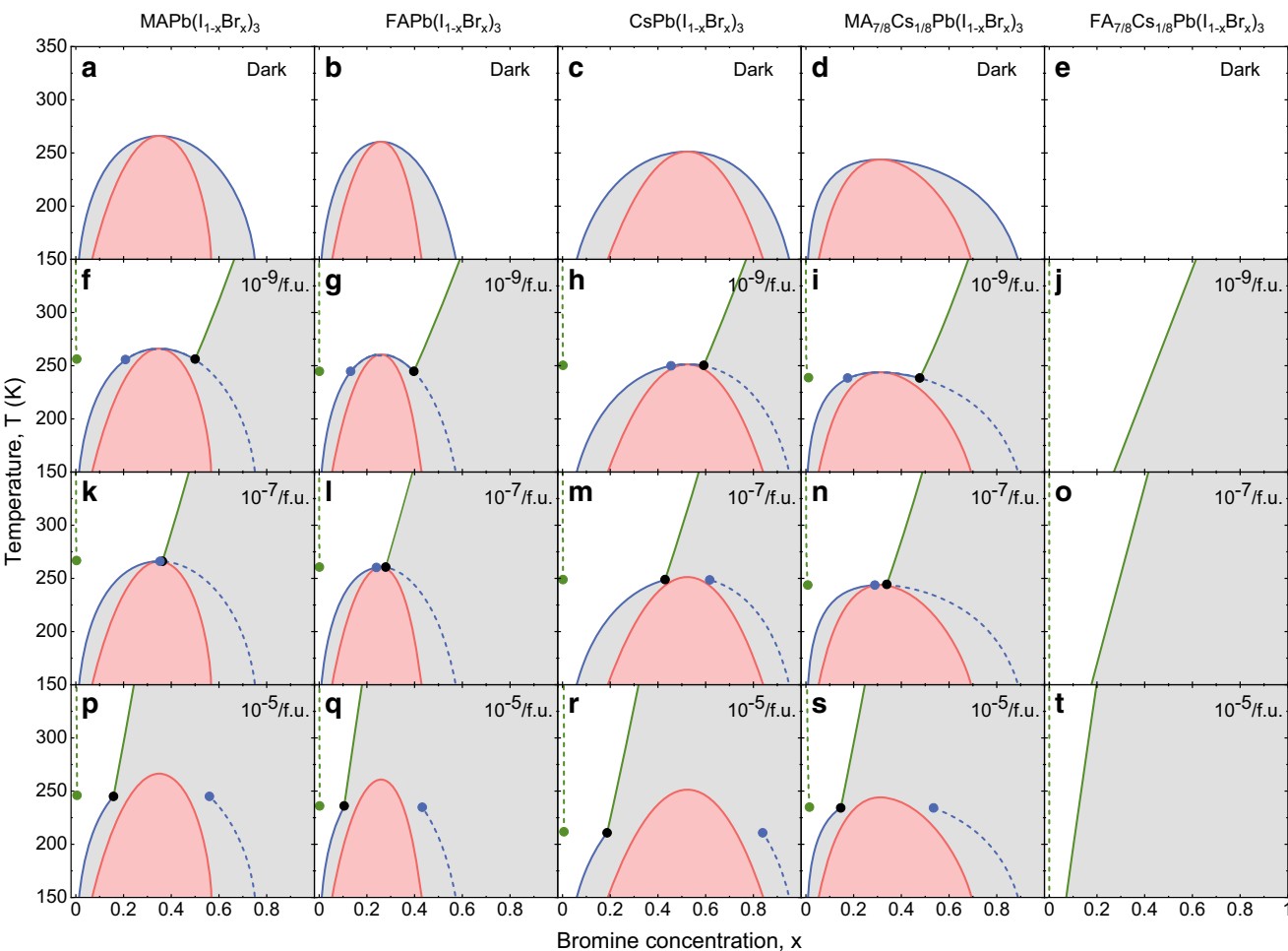

**Fig. 3 Phase diagrams at different photocarrier densities. a–e** Phase diagrams in the dark of the different compounds in the temperature window 150–350 K. **f–t** Phase diagrams for photocarrier densities $n = 10^{-9}$, $10^{-7}$, and $10^{-5}$ per formula unit. Red lines: spinodals separating the metastable (grey) and unstable (pink) regions. Full blue and green lines: binodals separating the stable (white) and metastable regions, with the blue (green) lines indicating the compositional (light-induced) binodals. When entering the metastable region by crossing the compositional (light-induced) binodals, nucleation of a phase with a Br concentration indicated by the dashed blue (green) lines becomes favourable. The dots indicate the possible coexistence of three phases: the parent phase (black dots) and two types of nucleated phases with different Br concentration (blue and green dots).

random orientations of the MA and FA molecules[44] can influence the phase diagrams. On the other hand, we expect that the observed trends in the phase diagrams are reliable, because the relative accuracies in the calculations of the free energies under I ↔ Br exchange for the investigated compounds are expected to be better than the absolute accuracies. It is important to note at this point that the threshold photocarrier densities for light-induced halide segregation (see next section) mainly depend on band gap differences and are hardly influenced by the phase diagrams in the dark.

Figure 3f–t show the phase diagrams of the five compounds under illumination, obtained from Eqs. (1–3) for different photocarrier densities in the mixed state of $n = 10^{-9}$, $10^{-7}$, and $10^{-5}$/f.u. Using an absorption coefficient $\alpha = 10^5\ \mathrm{cm}^{-1}$ and a photon energy $h\nu = 3\ \mathrm{eV}$[23], a value of $n = 5 \times 10^{-7}$/f.u. corresponds to an illumination intensity $I = 100\ \mathrm{mW\,cm}^{-2}$ of ~1 Sun ($n \approx I\alpha V\tau/h\nu$ when we neglect bimolecular recombination in the mixed state). We observe that the spinodals only slightly change with increasing photocarrier density. By contrast, important changes occur in the behaviour of the binodals with increasing $n$. Our theory predicts the existence of two types of binodals. The first type (full blue lines) can be viewed as a modification of the dark binodals by the illumination, which we will call

'compositional binodals'. Under illumination, a new type of binodals appears (full green lines), which we will call 'light-induced binodals'. These binodals can be crossed by increasing the illumination intensity, but also by decreasing the temperature. The latter is a prediction of our theory that is experimentally testable.

With increasing photocarrier density, the phase diagrams fall into two categories, observable for all compounds except $FA_{7/8}Cs_{1/8}Pb(I_{1-x}Br_x)_3$. (1) At low photocarrier density $n = 10^{-9}$/f.u. (Fig. 3f–i) the compositional binodals connect to the light-induced binodals after the critical point and both left and right branches of the binodals exist. When the left (right) branches of the compositional binodals are crossed by increasing (decreasing) $x$ or decreasing $T$, a phase is nucleated that is Br-richer (I-richer) than the parent phase, indicated by the dashed blue lines. We note that the dashed blue lines do not exactly coincide with the full blue lines at the top of the binodals. (2) At high photocarrier density $n = 10^{-5}$/f.u. (Fig. 3p–s) the compositional binodals connect to the green binodals before the critical point and only left branches of the compositional binodals exist. Figure 3k displays for $MAPb(I_{1-x}Br_x)_3$ a phase diagram that is very close to the transition between the two categories of phase diagrams, whereas Fig. 3l–n show both category-1 and category-2 phase

diagrams for $FAPb(I_{1-x}Br_x)_3$, $CsPb(I_{1-x}Br_x)_3$, and $MA_{7/8}Cs_{1/8}Pb(I_{1-x}Br_x)_3$.

For both categories of phase diagrams, a phase is nucleated that is I-richer than the parent phase when the light-induced binodals are crossed by increasing $x$ or decreasing $T$, as indicated by the green dashed lines. Interestingly, triple points $(x_{tr}, T_{tr})$ exist where two different phases with different halide composition can be nucleated from the parent phase. The Br concentrations of the parent phase $(x_{tr})$ and the two phases that can be nucleated at the triple points are indicated by dots in Fig. 3. For category-1 phase diagrams, the Br concentrations of the nucleated phases at the triple points (blue and green dots) both have a lower Br concentration than the parent phase (black dot), whereas for the category-2 phase diagram one nucleated phase is Br-richer and the other is I-richer than the parent phase. The predictions of two categories of phase diagrams and the existence of triple points are unique features of our theory. Their experimental observation by careful experimentation would be extremely interesting and could substantially increase our knowledge of light-induced halide segregation. $MAPb(I_{1-x}Br_x)_3$ could be a good candidate to experimentally investigate the occurrence of triple points. It is predicted by us to have the highest critical temperature (266 K) of the investigated compounds. This has the advantage that the thermally activated motion of the halide ions is the least suppressed around the critical point, which facilitates the observation of the segregation. Down to 235 K, for which $MAPbBr_3$ shows a cubic to tetragonal transition[45], no interfering structural transitions are expected. One can take $x$ slightly higher or slightly lower than the critical Br concentration $x_c = 0.35$ to investigate the triple points of the type shown in Fig. 3f and p, respectively. By tuning the temperature and the illumination level these triple points can then be searched for by looking at, e.g. different features in the absorption spectrum.

Our finding that $x \approx 0$ for the photosegregated I-rich phase (see the dashed green lines in Fig. 3) seems at odds with the experimental finding of Hoke et al. that $x \approx 0.2$ when segregation is complete[22]. An explanation for the latter finding was given by Ruth et al.[46]. In their kinetic Monte Carlo simulations of vacancy-mediated hopping of I and Br ions during the phase segregation process, Br ions get kinetically trapped in the I-rich nuclei, with a final concentration close to 0.2. Our theory applies to the onset of phase segregation and is therefore not incompatible with this result. There is recent experimental evidence from photoluminescence measurements that halide segregation commences with an almost I-pure phase, which gradually becomes less pure by the inclusion of Br[47,48]. This is in line with our argument.

**Threshold photocarrier densities**. Figure 4 shows for the five compounds results for the threshold photocarrier density $n_t$ for halide segregation. This is the value of $n$ at which the light-induced binodals are crossed for a given Br concentration $x$ and temperature $T$ (the full green lines in Fig. 3f–t). The light-induced nucleated phase is almost 100% I-rich (see the dashed green lines in Fig. 3f–t). From this fact, the following very accurate expression can be derived for $n_t$ (see 'Methods'):

$$n_t \approx f(x, T) \exp(-\Delta E_g(x)/k_B T), \quad (4)$$

where $\Delta E_g(x) \equiv E_g(x) - E_g(0)$. The prefactor in this expression is $f(x, T) \equiv \sqrt{(-\Delta F(x, T) + x\partial_x \Delta F(x, T)) V / k\tau E_g(x)}$. Equation (4) predicts extremely low thresholds $n_t$ at room temperature. We note that $n_t$ is the threshold photocarrier density in the mixed phase or in the parent phase at the onset of phase separation. The photocarrier density in the nucleated phase is according to Eq. (1) much larger. For example, for $MAPb(I_{0.5}Br_{0.5})_3$ we have $\Delta E_g(x = 0.5) \approx 0.28$ eV (see the green line in Fig. 2b), so that the

photocarrier density in the almost I-pure nucleated phase is at room temperature a factor of about $7 \times 10^4$ larger. This also means that, while the bimolecular recombination in the mixed or in the parent phase is negligible, this is definitely not the case in the nucleated phase.

In Fig. 4a we show $n_t$ as a function of Br concentration $x$ at room temperature. Apart from the extremely low $n_t$, an extremely strong dependence on $x$ is found. The threshold $n_t$ increases for different cation composition in the order FA–$FA_{7/8}Cs_{1/8}$–MA–$MA_{7/8}Cs_{1/8}$–Cs, which is the same order as the decrease in the steepness of the band gap curves in Fig. 2b. This order and the extremely strong dependence of $n_t$ on $x$ can be explained from the exponential factor $\exp(-\Delta E_g(x)/k_B T)$ in Eq. (4). We thus come to the important conclusion that the threshold photocarrier density is governed by the band gap difference of the mixed halide compound and the I-pure compound.

In Fig. 4b we show the $T$-dependence of $n_t$ at Br concentration $x = 0.5$ and in Fig. 4c the $T$-dependence of $x$ at a low, $n_t = 10^{-9}$/f.u., and a high, $n_t = 10^{-5}$/f.u., threshold photocarrier density. Both figures cover the temperature interval 300–350 K, which is a relevant operational range for solar cells. All curves in Fig. 4b are below the photocarrier density $n = 5 \times 10^{-7}$/f.u. at 1 Sun (dashed horizontal line), showing that all compounds with equal amounts of Br and I segregate at 1 Sun illumination. The steepness of the curves in Fig. 4b follows the same order as the steepness of the band gap curves in Fig. 2b. For both values of $n_t$ the curves in Fig. 4c show a decrease in steepness for the different compounds with an order that is opposite to the order in which the steepness of the band gap curves decreases. At the same time, the steepness of the curves for all compounds increases approximately proportionally to the logarithm of $n_t$. All these observations can be explained from the exponential factor $\exp(-\Delta E_g(x)/k_B T)$ in Eq. (4), although the factor $f(x, T)$ also contributes somewhat to the $T$-dependence.

Our results for the photostability of the different compounds agree with experimentally observed trends. First, we find that $CsPb(I_{1-x}Br_x)_3$ is more photostable than $MAPb(I_{1-x}Br_x)_3$. This is consistent with the experimental observations that $MAPb(I_{1-x}Br_x)_3$ is found to segregate for bromine concentrations $0.2 < x < 1$[22,40], while $CsPb(I_{1-x}Br_x)_3$ shows a smaller instability range $0.4 < x < 1$[17]. Second, we find that partial Cs alloying improves photostability, which is in agreement with the observed enhanced photostability by partial substitution of the organic cation in $MAPb(I_{1-x}Br_x)_3$ and $FAPb(I_{1-x}Br_x)_3$ by Cs[6,28,30–34]. This enhanced photostability is a direct consequence of the reduced dependence of the band gap on the Br concentration $x$ in Fig. 2b. To investigate if the trend of increasing stability with Cs loading pursues, we show in Supplementary Fig. 2d–f in Supplementary Note 4 phase diagrams in the dark and for photocarrier densities $n = 10^{-9}$ and $10^{-7}$/f.u. of $MA_{3/4}Cs_{1/4}Pb(I_{1-x}Br_x)_3$, where the Cs loading is 25% instead of the 12.5% loading in $MA_{7/8}Cs_{1/8}Pb(I_{1-x}Br_x)_3$. The phase diagrams are very similar to the latter compound, but are shifted down in temperature. The critical temperature decreases from 266 K without Cs loading ($MAPb(I_{1-x}Br_x)_3$), to 244 K for 12.5% loading, and 216 K for 25% loading, indeed showing a trend of increasing stability. This trend should be broken when increasing the Cs loading further, because for 100% Cs loading ($CsPb(I_{1-x}Br_x)_3$) the critical temperature is 250 K (see Fig. 3c). Supplementary Fig. 2g–i in Supplementary Note 4 show plots equivalent to Fig. 4a–c for $MA_{3/4}Cs_{1/4}Pb(I_{1-x}Br_x)_3$. The photocarrier density threshold is slightly higher than that of $MA_{7/8}Cs_{1/8}Pb(I_{1-x}Br_x)_3$, confirming the trend of increased photostability with increasing Cs loading.

Using $MAPb(I_{0.5}Br_{0.5})_3$ as an example, we predict at 300 K a photocarrier density threshold of $n_t = 9 \times 10^{-9}$/f.u. (see the green

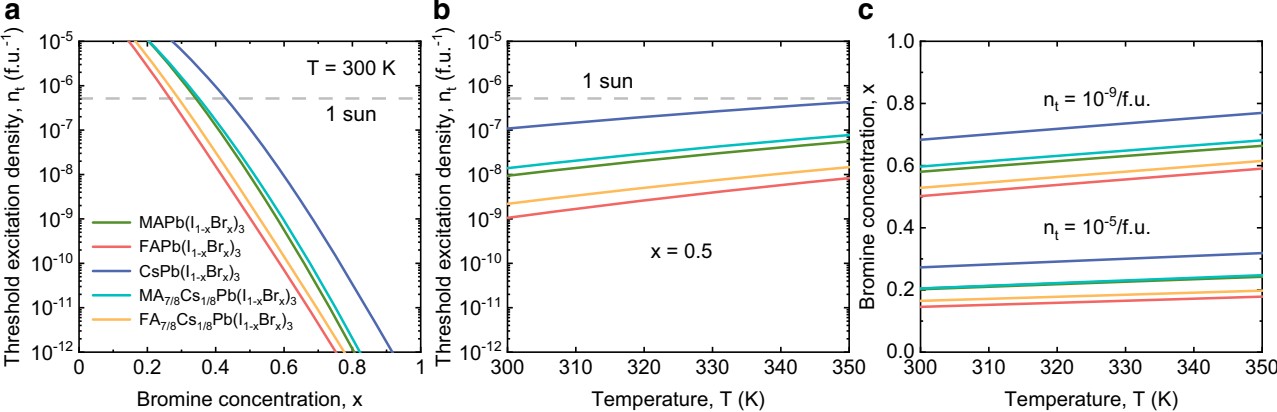

**Fig. 4 Threshold photocarrier density for light-induced halide segregation. a** Threshold photocarrier density $n_t$ for the different compounds at room temperature $T = 300$ K for light-induced halide segregation, as a function of Br concentration $x$. **b** Threshold photocarrier density as a function of temperature for $x = 0.5$. The horizontal dashed line in (**a**) and (**b**) indicates the photocarrier density $n = 5 \times 10^{-7}$/f.u. in the mixed phase for about 1 Sun illumination. **c** Bromine concentration as a function of temperature at threshold photocarrier densities $n_t = 10^{-9}$ and $10^{-5}$/f.u.

line in Fig. 4b), corresponding to an illumination threshold of 1750 $\mu$W cm$^{-2}$. In the recent experiments of ref. [25] a threshold for MAPb(I$_{0.5}$Br$_{0.5}$)$_3$ of about 35 $\mu$W cm$^{-2}$ is reported. We also predict an increase of the illumination threshold by a factor of 5.5 between 300 and 350 K, as compared to the experimentally found increase by a factor of about 3[25]. Considering the extremely strong sensitivity of $n_t$ to the various quantities in our theory, we find the agreement satisfactory. We note in particular the extremely strong dependency of $n_t$ on the Br concentration $x$ (see Fig. 4a), which leads to an extremely large sensitivity of the illumination threshold to preparation details of the perovskite film. The model of Ruth et al.[46] for the illumination threshold used in ref. [25] has a linear dependence on temperature and therefore predicts an increase of only about 17% between 300 and 350 K. We note that in this model the carrier diffusion lengths of electrons and holes, and a 'geometrical band gap volume' appear as parameters, quantities that do not appear in our theory.

It was recently reported that MAPb(I$_{0.2}$Br$_{0.8}$)$_3$ remixes for an illumination intensity of 200 W cm$^{-2}$[37]. This is equivalent to about 2000 Sun, which corresponds to a photocarrier density in the parent phase of about $10^{-3}$/f.u. Using Eq. (1) and the band gap difference between $x_1 = 0.8$ (parent phase) and $x_2 = 0$ (nucleated phase), we would find at room temperature a photocarrier density in a potentially nucleated phases of about $1.2 \times 10^6$/f.u. Obviously, our theory can no longer be applied at such extremely high densities, causing, among other things, a breakdown of the Boltzmann approximation used in Eq. (1). The explanation for the remixing could be that at these extreme densities there will be such a large spillover of photocarriers from the nucleated phase into the parent phase that the driving force for halide segregation disappears. This will essentially restore the conditions in the dark, where the mixed situation is favoured.

We remark that our theory has no adjustable parameters and can be the starting point for the inclusion of effects that were not yet accounted for. One such effect could be that the number of defects in the nucleated phase is larger than in the parent phase, possibly caused by a lattice mismatch between the two phases. This could lead to a lower photocarrier lifetime $\tau$ in the nucleated phase, which presently is assumed to be the same as in the parent phase. A lower $\tau$ in the nucleated phase will lead to a lower $n_t$. Another effect that will lower $n_t$ is a lattice compression of the I-rich nuclei by the surrounding mixed parent phase with smaller lattice constant, leading to a lowering of the band gap in the nucleated phase. If we take for MAPbI$_3$ under compressional strain the MAPb(I$_{0.5}$Br$_{0.5}$)$_3$ lattice constant instead of its relaxed

lattice constant, we find that the calculated DFT band gap is decreased by 0.145 eV. Using Eq. (4), this implies a decrease in $n_t$ by a factor of about 300 at room temperature and thus a six times lower illumination threshold than in the experiment of ref. [25]. In reality, the lattice adjustment to the surrounding phase will not be complete, so that a refined analysis may lead to a result in closer agreement with the experiment.

The consequence of the mechanism for light-induced halide segregation studied here is that the attractive band gap tunability of mixed halide perovskites at the same time leads to photostability problems. Nevertheless, routes towards optimal solutions follow from our study. For example, Fig. 4a shows that at 1 Sun illumination and room temperature CsPb(I$_{1-x}$Br$_x$)$_3$ should be photostable up to 42% Br concentration. This allows, according to Fig. 2b, reaching a band gap of 1.94 eV. This is more than sufficient for the top layer in an efficient tandem solar cell, which has an optimal band gap of 0.96 eV for the bottom and 1.63 eV for the top cell. For MAPb(I$_{1-x}$Br$_x$)$_3$ and MA$_{7/8}$Cs$_{1/8}$Pb(I$_{1-x}$Br$_x$)$_3$, Br concentrations of about 33% and 35% can be reached, allowing band gaps of 1.73 and 1.78 eV, respectively, which are both still sufficient.

## Discussion

We have presented a unified thermodynamic theory for light-induced halide segregation in mixed halide perovskites. The theory is based on minimization of the sum of a compositional free energy, obtained from binary alloying theory, and an electronic free energy of photocarriers, which distribute thermally over a nucleated phase and a parent phase with different band gaps due to different I-Br compositions. We applied the theory to MAPb(I$_{1-x}$Br$_x$)$_3$, FAPb(I$_{1-x}$Br$_x$)$_3$, CsPb(I$_{1-x}$Br$_x$)$_3$, as well as the partial Cs compounds MA$_{7/8}$Cs$_{1/8}$Pb(I$_{1-x}$Br$_x$)$_3$ and FA$_{7/8}$Cs$_{1/8}$Pb (I$_{1-x}$Br$_x$)$_3$. The spinodals in the Br concentration-temperature, $x$-$T$, phase space, separating unstable and metastable regions, only slightly change for photocarrier densities corresponding to relevant illumination intensities. In addition to compositional binodals that are also present in the dark, new light-induced binodals appear, signalling the nucleation of an I-rich phase from the parent phase. These binodals, which are attributed to funneling of photocarriers into the low-band gap I-rich phase, occur at an extremely small photocarrier density and illumination intensity governed by the band gap difference between the mixed phase and the nucleated I-rich phase.

Several predictions of the theory are in agreement with experimental findings, such as a strongly temperature and composition dependent illumination threshold for halide segregation

and a stabilization effect upon alloying FA or MA with Cs. The fundamental reason for this stabilization effect is that mixing Cs into FA or MA reduces the unit cell volume, leading to smaller band gap differences between the parent and I rich phases. The theory predicts two categories of phase diagrams and the existence of photocarrier density-dependent triple points $(x_{tr}, T_{tr})$ below room temperature, where two phases with different Br-I compositions can be nucleated from the parent phase. The experimental study of these novel physical phenomena would be extremely interesting and increase our understanding of light-induced halide segregation. The theory is flexible and transferable, making it a suitable starting point for refinements to include effects that have not yet been considered, such as different recombination rates in the different phases and changes in the band gap due to strain. The theory can also be readily applied to other semiconductors where the band gap is tuned by alloying.

We finally note that a metastable region in phase space is entered when the illumination intensity exceeds the threshold for halide segregation. According to nucleation theory, a surface free energy due to the presence of an interface between the nucleated and parent phase could inhibit phase separation. Phase separation then requires the crossing of a free energy barrier composed of a positive surface free energy and the negative bulk free energy for a nucleus of a critical size. For the investigation of a phase separation inhibition effect, it would be important to evaluate the surface free energy between a mixed I-Br and an I-rich phase, and the probability that a nucleus will grow spontaneously to a critical size. Differences in surface free energy and sizes of critical nuclei for different perovskites could provide additional handles to suppress light-induced halide segregation. We suggest that the delayed onset for the acceleration of the segregation reported in ref. [38] is related to the induction time for the formation of critical nuclei. The existence of such an induction time is a well-known phenomenon in nucleation theory[49].

## Methods

**Calculation of total energies.** To calculate the total energies of the single-cation mixed halide perovskites MAPb(I$_{1−x}$Br$_x$)$_3$, FAPb(I$_{1−x}$Br$_x$)$_3$, and CsPb(I$_{1−x}$Br$_x$)$_3$, we start from a periodic supercell of the pure I compounds containing 2 formula units, with a $2 \times 1 \times 1$ expansion of a (pseudo)cubic perovskite building block. We then replace I anions by Br anions at different concentrations $x = 0$, 1/6, 1/3, 1/2, 2/3, 5/6, 1. The total number of possible configurations for each single-cation perovskite is $2^6 = 64$. For the Cs perovskite with perfect $O_h$ symmetry the three halide sites are equivalent, which reduces the total number of inequivalent configurations to 21. Accounting for the deviation from $O_h$ symmetry in the case of the MA and FA perovskites leads to an increase to 36 inequivalent configurations. To include the Cs cations in the double-cation perovskites MA$_{7/8}$Cs$_{1/8}$Pb(I$_{1−x}$Br$_x$)$_3$ and FA$_{7/8}$Cs$_{1/8}$Pb(I$_{1−x}$Br$_x$)$_3$, the $2 \times 1 \times 1$ supercells of the 36 inequivalent MAPb(I$_{1−x}$Br$_x$)$_3$ and FAPb(I$_{1−x}$Br$_x$)$_3$ configurations are repeated in two directions to construct $2 \times 2 \times 2$ supercells containing 8 formula units. The double-cation perovskites are then constructed by substituting one of the 8 organic cations by Cs. The total number of inequivalent configurations is then $2 \times 36 = 72$, where factor 2 reflects the two inequivalent Cs substitutions in the $2 \times 2 \times 2$ supercell. In the case of MA$_{3/4}$Cs$_{1/4}$Pb(I$_{1−x}$Br$_x$)$_3$, studied in Supplementary Note 4, we substitute two of the MA cations on the body diagonal of the supercell by Cs cations, preserving in this way symmetry as much as possible.

The total energy calculations are performed within Density-Functional Theory (DFT). We use the projected augmented wave (PAW)[50] method and the Perdew-Burke-Ernzerhof exchange-correlation functional revised for solids (PBEsol)[51] within the generalized gradient approximation (GGA)[52], as implemented in the Vienna ab initio simulation package (VASP)[53]. We use $4 \times 8 \times 8$ and $4 \times 4 \times 4$ $k$-point Brillouin zone samplings for the single-cation and double-cation compounds, respectively, and a plane-wave kinetic energy cutoff of 500 eV. The shape and volume of the unit cell as well as the atomic positions in the unit cell of each configuration are fully optimized. The energy and force convergence parameters are set at 0.01 meV and 0.005 eV/Å, respectively.

**Calculation of the mixing free energy.** The mixing enthalpies $\Delta U_j$ per formula unit of the inequivalent configurations $j = 1, \dots J$ with relative Br concentration $x$ are given by

$$\Delta U_j = E_j - (1-x)E_{APbI_3} - xE_{APbBr_3}, \qquad (5)$$

where $E_j$, $E_{APbI_3}$, and $E_{APbBr_3}$ are the total energies per formula unit of configuration $j$, the pure I configuration, and the pure Br configuration, respectively. 'A' denotes MA, FA, Cs, MA$_{7/8}$Cs$_{1/8}$, or FA$_{7/8}$Cs$_{1/8}$. The resulting enthalpies are given by the $J = 36$ points in Fig. 1f, g for MAPb(I$_{1−x}$Br$_x$)$_3$ and FAPb(I$_{1−x}$Br$_x$)$_3$, respectively, and the $J = 21$ points in Fig. 1h for CsPb(I$_{1−x}$Br$_x$)$_3$. To treat the double-cation perovskites on the same footing as the single-cation perovskites we take the average of the total energies of the two inequivalent Cs substitutions, resulting in enthalpies given by the $J = 36$ points in Fig. 1i, j. We checked that the mixing free energy curves (see below) for the two inequivalent Cs substitutions are almost indistinguishable, which validates taking this average.

We apply the quasi-chemical approximation (QCA)[41] to obtain the mixing enthalpy $\Delta U(x, T)$, entropy $\Delta S(x, T)$, and Helmholtz free energy $\Delta F(x, T)$ as functions of the Br concentration $x$ and temperature $T$. The QCA has been successfully employed in the thermodynamic analysis of semiconductor alloys[42,54,55]. In the QCA, the perovskite lattice is decomposed into microclusters, which are treated independently[41]. For the different microclusters we take the $J$ inequivalent configurations mentioned above. In the QCA, the thermal equilibrium fraction of microclusters with configuration $j$ is given by

$$\bar{x}_j = \frac{g_j \exp\left(\left(n_j(Br)\mu_{Br} - \Delta U_j\right)/k_B T\right)}{\sum_j g_j \exp\left(\left(n_j(Br)\mu_{Br} - \Delta U_j\right)/k_B T\right)}, \qquad (6)$$

where $g_j$ is the degeneracy and $n_j(Br) = 0, 1, 2, 3$ the number of Br anions per formula unit of the configuration, and $\mu_{Br}$ is the Br chemical potential. This chemical potential should be determined such that the desired Br concentration is obtained:

$$\sum_j n_j(Br)\bar{x}_j = 3x. \qquad (7)$$

After having found $\mu_{Br}$ from this condition, the mixing enthalpy and entropy per formula unit are obtained as:

$$\Delta U(x, T) = \sum_j \bar{x}_j \Delta U_j, \qquad (8)$$

and

$$\Delta S(x, T) = -3k_B \left[(1-x)\ln(1-x) + x\ln x + \sum_j \bar{x}_j \ln(\bar{x}_j/x_j^0)\right], \qquad (9)$$

with $x_j^0 = g_j x^{n_j(Br)}(1-x)^{3-n_j(Br)}$. The mixing free energy is finally obtained as $\Delta F(x, T) = \Delta U(x, T) - T\Delta S(x, T)$.

**Determination of the binodals and spinodals.** To find the binodals we consider the possibility, starting from the mixed situation with Br concentration $x$, to decrease the free energy by demixing through nucleation of a phase with a concentration $x_2 \neq x$ with a small volume fraction $\delta\phi \equiv \phi_2$. In the mixed situation we can make the simplifying approximation $n \approx G/\tau$ for the photocarrier density, because for all relevant illumination intensities bimolecular recombination is then negligible. The free energy Eq. (2) in the mixed situation is $\Delta F^\star(x, x, 1, 0, T)$, while the free energy in the demixed situation is $\Delta F^\star(x_1, x_2, 1 - \delta\phi, \delta\phi, T)$, with, to linear order in $\delta\phi$,

$$x_1 = x - (x_2 - x)\delta\phi. \qquad (10)$$

The difference in free energy between the demixed and mixed situations is then to linear order in $\delta\phi$:

$$\delta\Delta F^\star = \delta\phi\Big\{\Delta F(x_2, T) - \Delta F(x, T) - (x_2 - x)\partial_x\Delta F(x, T) \\ + n\left[\exp\left(-\frac{E_g(x_2) - E_g(x)}{k_B T}\right)\left(E_g(x_2) - E_g(x)\right) - (x_2 - x)E_g'(x)\right] \\ - n^2 k\tau \exp\left(-2\frac{E_g(x_2) - E_g(x)}{k_B T}\right)E_g(x)/V\Big\}, \qquad (11)$$

with $\partial_x\Delta F(x, T) \equiv \partial\Delta F(x, T)/\partial x$. When $\delta\Delta F^\star < 0$, the demixed situation has lower free energy than the mixed situation. We thus find the binodals in Fig. 3 for a certain photocarrier density $n$ by looking in $x$-$T$ phase space for a value of $x_2$ of a nucleated phase for which $\delta\Delta F^\star = 0$. The dashed lines in Fig. 3 give the Br concentration $x_2$ of the nucleated phase. For the light-induced binodals $x_2$ is very small, indicating the nucleation of a low-band gap I-rich phase. When we put $x_2 = 0$ in Eq. (11) we get

$$\delta\Delta F^\star \approx \delta\phi\Big\{-\Delta F(x, T) + x\partial_x\Delta F(x, T) + n\left[-\Delta E_g(x)\exp\left(\frac{\Delta E_g(x)}{k_B T}\right) + xE_g'(x)\right] \\ - n^2 k\tau \exp\left(2\frac{\Delta E_g(x)}{k_B T}\right)E_g(x)/V\Big\}, \qquad (12)$$

where $\Delta E_g(x) = E_g(x) - E_g(0)$ is the band gap difference between the mixed and the pure I phase. The term $xE_g'(x)$ is found to be very small and can be neglected. Putting $\delta\Delta F^\star = 0$ and solving for $n$ yields the threshold photocarrier density Eq. (4), which provides an extremely good approximation to the curves in Fig. 4.

To find the spinodals we consider the possibility to decrease the free energy by generating a volume fraction $\phi$ of a phase with a slightly different concentration $x_2 = x + \delta x$. The free energies in the demixed situation can now be written as $\Delta F^\star(x - \phi \delta x/(1 - \phi), x + \delta x, 1 - \phi, \phi, T)$. To second order in $\delta x$ the difference in free energy then becomes

$$\delta \Delta F^\star = \frac{\phi(\delta x)^2}{2(1 - \phi)} \left\{ \partial_x^2 \Delta F(x, T) + n \left[ -2 \frac{\left( E_g'(x) \right)^2}{k_B T} + E_g''(x) \right] \right\}. \quad (13)$$

When $\delta \Delta F^\star < 0$, the demixed situation can be established from the mixed situation in a continuous way, without crossing a free energy barrier. Putting $\delta \Delta F^\star = 0$ thus yields the spinodal separating the metastable from the unstable region in $x$-$T$ phase space.

We note that for the unilluminated case ($n = 0$, see the top panels in Fig. 3) the above procedures to find the binodals and spinodals are identical to the usual procedures, where the binodals are found from a common tangent construction and the spinodals from the inflection points of $\Delta F$ as a function of $x$. These usual procedures can be applied when the mixing free energy is equal to the volume-weighted sum of the free energies per volume of the different phases, which holds in the dark but not under illumination.

## Data availability

Data supporting this publication are available from the corresponding author on request. The calculated volumes of the unit cells of the perovskites studied in this work are given in Supplementary Fig. 1. The used formulas for the band gaps of the perovksites are given in Supplementary Table 1.

## Code availability

The DFT calculations were performed with the code VASP ('Vienna Ab initio Simulation Package'), available at http://www.vasp.at.

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

## Acknowledgements

The authors thank Prof. Dr. René Janssen and MSc Kunal Datta for stimulating discussions about the subject of this work. ZC acknowledges funding from the Eindhoven University of Technology. ST acknowledges funding by the Computational Sciences for Energy Research (CSER) tenure track program of Shell and NWO (Project No. 15CST04-2) as well as NWO START-UP from the Netherlands.

## Author contributions

The project was conceived and planned by P.A.B. and S.T. All calculations were done by Z.C. S.T. and G.B. guided the work on the DFT calculations and P.A.B. guided the work on the phase diagrams. The first version of the paper was written by Z.C. under the guidance of P.A.B. All authors contributed to the results of this work and to the final version of the paper.

## Competing interests

The authors declare no competing interests.
