## [Peer Review File · Nature Communications]

REVIEWER COMMENTS

Reviewer #1 (Remarks to the Author):

This is a theoretical study of halide photosegregation in mixed halide perovskite materials. The authors develop a model applicable to five different mixed halide perovskite systems wherein the distinguishing feature of the model is the use of photocarrier energies to estimate free energies under illumination. The reviewer identifies the following as key novelty points of the presented work (a) a theoretical study of photosegregation in five different cation perovskites and (b) the establishment of temperature-dependent excitation intensity thresholds to halide photosegregation. Although the authors suggest that the use of photocarrier energies to estimate mixing free energies under illumination is a novelty point, the reviewer notes that Reference 23 already discusses this concept. Despite this, the reviewer found the work interesting and potentially useful as there are predictions in the model that can (potentially) be tested. The reviewer therefore recommends publication after the authors address the following comments.

Comments

1. Please comment on Knight's work (Electronic Traps and Phase Segregation in Lead Mixed-Halide Perovskite, A. J. Knight et al. ACS Energy Lett. 2019, 4, 1, 75–84) where they report the existence of an induction time for halide segregation within the context of the author's threshold excitation intensity for photosegregation.

2. The authors write that a prior model for halide photosegregation based on band gap differences does not account for a temperature dependence. The reviewer notes that a follow up paper to Reference 23 by Ruth (Vacancy-mediated anion photosegregation kinetics in mixed halide hybrid perovskites: Coupled kinetic Monte Carlo and optical measurements, A. Ruth et al. ACS Energy Lett. 2018, 3, 2321-2328.) does contain a temperature dependence to the threshold excitation intensity. A comparison of the predicted temperature dependencies between those of the authors and those of Ruth is therefore warranted for MAPb(I_{1-x}Br_x)₃.

3. The authors use DFT structures that contain fully-aligned MA or FA molecules when discussing X-site mixing. However, it has been demonstrated that structures which start from randomly oriented MA or FA molecules pick up key details of A-site alloying which calculations containing fully-aligned molecules do not. (Formation and Composition-Dependent Properties of Alloys of Cubic Halide Perovskites, G. M. Dalpian et al. Chem. Mater. 2019, 31, 7, 2497–2506). How confident can we be in the mixing enthalpies given this difference?

4. A key point of the model is that partial replacement of organic cations by Cs reduces band gap difference between parent and I-rich phases. It is known that the A-site cation does not directly contribute to the band gap of perovskites. Consequently, it would seem that partial substitution of A-site cations with Cs would alter the band gap of both parent and I-rich regions to a similar extent. How is this accounted for in the model?

5. The authors use Vegard's law for halide-composition dependent band gaps. Formulas for each studied composition are in Table SI. Above the table, band gap values used to estimate/calculate corresponding mixed-cation band gaps are provided. The reviewer points out that the values of single cation systems provided, based on Ref S6, are different from those used in the Vegard's law equations in Table SI. This discrepancy should be corrected.

6. The reviewer would like to see estimates for the photosegregated fraction in the various materials. It has previously been estimated through experiment that segregated fractions represent a minority of the overall composition of the mixed halide material, taking values of order ~0.1-0.2. Consequently, phase fraction estimates at different temperatures and light intensities would be useful.

7. Figures 3a-e presents phase diagrams for all studied cation compositions. While all phase diagrams qualitatively resemble each other, the FACs phase diagram is surprisingly different. This should be discussed more explicitly.

8. The reviewer would like the authors to more explicitly clarify the discussion in Figure 3 regarding stable and unstable compositions and the effects of photosegregation. An examination and interpretation of the discussion regarding the solid green lines in Figure 3 would suggest the following: (a) photosegregation leads to universal terminal $x \sim 0$ for all unstable stoichiometries. These are indicated by the dashed vertical green lines, (b) stable stoichiometries are temperature dependent. So whereas $\text{MAPb}(\text{I}_{1-x}\text{Br}_x)_3$ $x \sim 0.57$ is stable at 10⁻⁹/f.u. at 300 K, it becomes unstable at $T < 300$ K under the same conditions. If correct, this should be stated more clearly as these are experimental testables of the model. Furthermore, the authors should comment on why a terminal $x \sim 0$ value at room temperature appears to differ from experimental terminal x values of $x \sim 0.2$ in $\text{MAPb}(\text{I}_{1-x}\text{Br}_x)_3$ (e.g. see Hoke et al. Chem. Sci., 2015, 6, 613) and analogous non-zero terminal x values in other systems. If kinetic constraints to reaching terminal x values of $x \sim 0$ are implicitly assumed then this should be stated in the text.

9. The authors predict the existence of triple points where 3 different phases would coexist. This prediction distinguishes this theory from other theories of photosegregation. Verifying the existence of these triple points through experiment would improve our understanding of photosegregation. However, these triple points are all predicted to be below room temperature where phase transitions may invalidate some assumptions of the model. The authors should therefore explicitly state which compositions are most-likely to have triple point that can be experimentally observed.

10. Following the authors' conclusions about band gap differences in mixed-halide alloys dictating the phase stability, the reviewer concludes that alloys having small band gap slopes in Vegard's law would be predicted to be more stable. i.e. $\text{CsPbI}_{1-x}\text{Br}_x$ should be the most stable out of all studied compositions. Following this line of thought, the reviewer would like the authors to simulate higher Cs loadings (e.g. 25%, 50%, 75%) into MA or FA based mixed halide perovskites to support their conclusions.

11. While the model findings suggest new insights into halide photosegregation, it would be best if these predictions were tested. The reviewer understands that the authors come from the modeling side. Consequently, it would seem that what could be done is for the authors to conduct an exhaustive literature search to find corroborating evidence for their model predictions.

12. It would be useful if the authors ended their study by explicitly suggesting what optimal (stable) compositions for each system are.

Reviewer #2 (Remarks to the Author):

This topic has attracted a lot of attention due to the need for stable mixed halide perovskites. The paper presents a parameter-free theory showing how A-site alloying in mixed halide perovskites influences the thermodynamic stability under illumination. Construction of the phase diagrams for each compound in the dark and under illumination, distinguishing stable, metastable, and unstable regions is clearly described. The theory can be refined to include effects such as different recombination rates in the different phases and changes in the band gap due to strain, giving the paper additional impact. The authors have shown how they have gone beyond the large number of papers already published on this topic. I therefore recommend publication in Nature Communications.

There is a clarification that would help the reader understand the model which I would like the authors to address.

At the bottom of page 5, it states 'The key ingredient of our unified theory for light-induced halide segregation is the consideration of the combination of the compositional free energy in the dark and the free energy of photocarriers in the presence of illumination' This sentence needs relating to the description on page 6 in the text and Figure 2a. Here, the authors argue that there is inward diffusion of I and outward diffusion of Br, which needs more information. I diffusion is driven by a lower concentration in the region towards which the I ions are diffusing, yet this region is I rich. Do we have to infer that an I ion that happens to reach an I rich region is captured due the region

having a lower bandgap? How is this effect linked to the Br fraction variation of the bandgap presented in Figure 2b? What is the effect on the free charge carriers? Why should there be a net outward diffusion of Br? What is the effect of the illumination in addition to generating free carriers? The extent that these questions are addressed in the following text should be discussed at this stage.

Reviewer #3 (Remarks to the Author):

Chen et al present a thermodynamic model to describe photoinduced halide segregation (PHS) in perovskite films with different compositions ('A'-cations, halides). The critical photon flux threshold for phase segregation to occur is linked to the bandgap difference between the segregated phases which form under illumination.

The manuscript is well written and addresses a field of general interest and a phenomenon which currently limits the full exploitation of the wavelength tunability of lead halide perovskites. I recommend that this work could be published in Nature Comm after addressing the following points:

- 1) Given that there are already a number of theories and models to explain PHS it would be great if the authors could point out more clearly what component of their theory has been described already and where the paper is breaking new ground.
- 2) What drives halide segregation in the first place when a perovskite crystal with homogeneously distributed halides is illuminated?
- 3) A 'unified theory' for PHS should also be able to explain the recent observation that halide segregation can be reversed at higher photon fluxes (Mao et al - Light-induced reversal of ion segregation in mixed-halide perovskites). How can this be achieved?
- 4) Rational for the effect of Cs in PHS: The authors should summarise the concepts introduced in reference 33 more clearly to explain the effect of Cs.
- 5) Page 11: the authors argue that 'The light-induced nucleated phase is almost 100% I-rich'. This is very different from the experimental results in the following 2 references: Chem. Sci. 6, 613–617 (2015) and DOI 10.1038/s41563-020-00826-y, where the stabilised segregated I-rich phases are not I-pure phase. This would require some more discussion.

REVIEWER COMMENTS

Reviewer #1 (Remarks to the Author):

This is a theoretical study of halide photosegregation in mixed halide perovskite materials. The authors develop a model applicable to five different mixed halide perovskite systems wherein the distinguishing feature of the model is the use of photocarrier energies to estimate free energies under illumination. The reviewer identifies the following as key novelty points of the presented work (a) a theoretical study of photosegregation in five different cation perovskites and (b) the establishment of temperature-dependent excitation intensity thresholds to halide photosegregation. Although the authors suggest that the use of photocarrier energies to estimate mixing free energies under illumination is a novelty point, the reviewer notes that Reference 23 already discusses this concept. Despite this, the reviewer found the work interesting and potentially useful as there are predictions in the model that can (potentially) be tested. The reviewer therefore recommends publication after the authors address the following comments.

We are pleased with the positive attitude of the reviewer to our work and agree with the reviewer about the key novelty points. We also agree that Ref. 23 (Draguta et al.) already discusses the use of photocarrier energies to estimate mixing free energies, as we acknowledge in the manuscript (first paragraph on p. 3: “A model based on band gap differences between perovskites with different halide composition does account for a threshold illumination intensity[23]. ...”). We are pleased that the reviewer recommends publication in Nature Communications. Below, we address the comments of the reviewer.

Comments

1. Please comment on Knight's work (Electronic Traps and Phase Segregation in Lead Mixed-Halide Perovskite, A. J. Knight et al. ACS Energy Lett. 2019, 4, 1, 75–84) where they report the existence of an induction time for halide segregation within the context of the author's threshold excitation intensity for photosegregation.

We thank the reviewer for drawing our attention to the phenomenon of induction time in this work (Ref. 38 in our manuscript). In this work, a delayed onset for halide segregation after illumination is reported. We assume that the delay time in that work is what the reviewer means by “induction time”. We think that the existence of an induction time (or delay time) is related to the formation of critical nuclei of the low-band gap phase. Phase segregation can only occur after critical nuclei are formed for which the increase in interface free energy is equal to the decrease in bulk free energy by the creation of the nucleated phase. As is well known in nucleation theory, the formation of a critical nucleus requires an induction time (see, for example the book of Kashchiew, new Ref. 49 in our revised manuscript). We suggest that this is the explanation for the observed delay time in light-induced segregation. We added this explanation to the last paragraph of the paper where we discuss the existence of critical nuclei. Of course, critical nuclei only exist above the threshold excitation intensity for photosegregation. All illumination experiments in Ref. 38 have been done well above this threshold.

The addition to the manuscript is

“We suggest that the delayed onset for the acceleration of the segregation reported in Ref. 38 is related to the induction time for the formation of critical nuclei. The existence of such an induction time is a well-known phenomenon in nucleation theory [49].”

2. The authors write that a prior model for halide photosegregation based on band gap differences does not account for a temperature dependence. The reviewer notes that a follow up paper to Reference 23 by Ruth (Vacancy-mediated anion photosegregation kinetics in mixed halide hybrid perovskites: Coupled kinetic Monte Carlo and optical measurements, A. Ruth et al. ACS Energy Lett. 2018, 3, 2321-2328.) does contain a temperature dependence to the threshold excitation intensity. A comparison of the predicted temperature dependencies between those of the authors and those of Ruth is therefore warranted for MAPb(I_{1-x}Br_x)₃.

Indeed, Ruth et al. propose in this paper a model for light-induced halide segregation that contains a temperature dependence of the threshold excitation density. This model is applied to their recent experimental work in Ref. 25 (Elmelund et al.). We want to point out the similarities and differences between our theory and the theory presented in Ruth et al. In both theories, the central idea is that photocarriers are funneled to an I-rich nucleated phase. Crucial differences are the dependence of the illumination threshold in Ruth et al. on the carrier diffusion lengths of electrons and holes and on a “geometrical band gap volume”, quantities that do not appear in our theory. The expression for the illumination threshold in Ruth et al. has a linear T-dependence and therefore predicts a threshold increase of only about 17% between 300 and 350 K, instead of the experimentally found increase by a factor of about 3 in Ref. 25. Our theory predicts a substantial threshold increase by a factor 5.5; see the green line in Fig. 4b and the fifth paragraph of the section “Threshold photocarrier densities” in the main text. Our result does deviate from experiment, but we suggest in the one-but-last paragraph of this section improvements of our theory that could yield a better agreement.

We have added the following discussion of the model of Ruth et al. to fifth paragraph of this section:

“The model of Ruth et al. [46] for the illumination threshold used in Ref. [25] has a linear dependence on temperature and therefore predicts an increase of only about 17% between 300 and 350 K. We note that in this model the carrier diffusion lengths of electrons and holes, and a “geometrical band gap volume” appear as parameters, quantities that do not appear in our theory.”

3. The authors use DFT structures that contain fully-aligned MA or FA molecules when discussing X-site mixing. However, it has been demonstrated that structures which start from randomly oriented MA or FA molecules pick up key details of A-site alloying which calculations containing fully-aligned molecules do not. (Formation and Composition-Dependent Properties of Alloys of Cubic Halide Perovskites, G. M. Dalpian et al. Chem. Mater. 2019, 31, 7, 2497–2506). How confident can we be in the mixing enthalpies given this difference?

The referee is right that this is a concern. We do think that our calculations of the phase diagrams in the dark improve on the calculations of Brivio et al. (Ref. 42), but acknowledge that there are uncertainties about our phase diagrams in the dark because of, among other things, randomly oriented MA or FA molecules. The fact that we are interested in energy differences under I \leftrightarrow Br replacement rather than absolute energies will take away some of these uncertainties. We also note that all structures we consider are fully relaxed, always allowing for reorientations of the MA and FA molecules (so, the MAs and FAs are in general not aligned). Furthermore, we are confident that our threshold photocarrier densities for light-induced halide segregation are quite reliable, because they mostly depend on band gap differences (Eq. (4)), which we take from experiment.

To be clearer about this, we added to the discussion of the work Brivio et al. in the second paragraph of the section “Phase diagrams” the following:

“This comparison does show that subtle differences in the way the free energy is calculated can have a substantial influence on the phase diagram. Therefore, also our phase diagrams can have inaccuracies that are related to, e.g., the specific exchange-correlation functional used in the DFT calculations (see “Methods”) and the limited size of the used supercells. Also, thermally induced random orientations of the MA and FA molecules [44] (Dalpian et al.) can influence the phase diagrams. On the other hand, we expect that the observed trends in the phase diagrams are reliable, because the relative accuracies in the calculations of the free energies under I \leftrightarrow Br exchange for the investigated compounds are expected to be better than the absolute accuracies. It is important to note at this point that the threshold photocarrier densities for light-induced halide segregation (see next section) mainly depend on band gap differences and are hardly influenced by the phase diagrams in the dark.”

4. A key point of the model is that partial replacement of organic cations by Cs reduces band gap difference between parent and I-rich phases. It is known that the A-site cation does not directly contribute to the band gap of perovskites. Consequently, it would seem that partial substitution of A-site cations with Cs would alter the band gap of both parent and I-rich regions to a similar extent. How is this accounted for in the model?

The A-site cation does indeed not directly contribute to the band gap of these perovskites. However, A-cation alloying can change the band gap indirectly by changing the volume of the ABX₃ unit cell or by introducing octahedral distortions, as remarked in the second paragraph of our manuscript on p.2 (Ref. 35, Tao et al.). These structural deformations change the hybridization between the B- and X-site ions, which changes the conduction band minimum and valence band maximum energies. For the double-cation compounds MA_{7/8}Cs_{1/8}Pb(I_{1-x}Br_x)₃ and FA_{7/8}Cs_{1/8}Pb(I_{1-x}Br_x)₃, the band gaps are obtained from an interpolation scheme (see Supplementary Note S2). In Fig. 2b, it is seen that the dependence of the band gap on the bromine concentration x for the double-cation compounds is less steep than for the single-cation compounds. This decrease in steepness in MA_{7/8}Cs_{1/8}Pb(I_{1-x}Br_x)₃ or FA_{7/8}Cs_{1/8}Pb(I_{1-x}Br_x)₃ will reduce the band gap difference between the parent and nucleated I-rich phase. This, according to Eq. (4), leads to a larger photocarrier density threshold for halide segregation and hence to greater photostability.

To make this clearer, we added to the second paragraph of the section “Light-induced halide segregation” the sentence

“The decrease in steepness of the band gap curves with Cs alloying may look surprising, because the A-site cation does not directly contribute to the states governing the band gap. However, A-cation alloying can change the band gap indirectly by changing the volume of the unit cell or by introducing octahedral distortions [35].”

and to the fourth paragraph of the section “Threshold photocarrier densities” the sentence

“This enhanced photostability is a direct consequence of the reduced dependence of the band gap on the Br concentration x in Fig. 2b.”

5. The authors use Vegard’s law for halide-composition dependent band gaps. Formulas for each studied composition are in Table SI. Above the table, band gap values used to estimate/calculate corresponding mixed-cation band gaps are provided. The reviewer points out that the values of single cation systems provided, based on Ref S6, are different from those used in the Vegard’s law equations in Table SI. This discrepancy should be corrected.

This is indeed a slight inconsistency. Instead of using the values from calculations (Tao et al.) in Supplementary Note S2, we now use the experimental values, which are slightly different.

We have recalculated the phase diagrams of the double-cation compounds. The difference is minor.

6. The reviewer would like to see estimates for the photosegregated fraction in the various materials. It has previously been estimated through experiment that segregated fractions represent a minority of the overall composition of the mixed halide material, taking values of order $\sim 0.1-0.2$. Consequently, phase fraction estimates at different temperatures and light intensities would be useful.

We agree that it would be useful to have estimates of the final photosegregated fraction at different temperatures and light intensities. Extensions of our theory beyond the present work are needed for this. The ingredients to be added are the following. 1) Our present theory is limited to the initial stage of halide segregation, where the carrier diffusion lengths do not play a role (see Supplementary Note SI3). Extension of the theory to describe the complete photosegregation process would require explicit inclusion of the carrier diffusion lengths. 2) The final compositions and fractions of the phases may depend on kinetic effects. For example, it is expected that bromine ions get kinetically trapped in the iodine-rich phase during the growth of this phase (see our reply to point 8 of the reviewer). Finding the final phase compositions and fractions will then require consideration of the kinetics of the halide segregation, which is outside the scope of our present thermodynamic theory. In absence of these ingredients, our theory can provide information about the onset of photosegregation but not yet about final phase compositions and fractions.

To explore the possibilities to obtain the final phase compositions and fractions from our theory we performed test calculations for $\text{MAPb}(\text{I}_{0.5}\text{Br}_{0.5})_3$ based on the following idea. In our theory, photosegregation will stop when the photocarrier density in the parent phase from which nucleation takes place is below the threshold density. From our theory, the threshold density for a bromine concentration x_1 in the parent phase at that moment can be found. From experiment we know that the bromine concentration in the photosegregated phase is $x_2 \approx 0.2$ (see point 8 below; we think that this concentration is kinetically determined). Eqs. (1) and (2) in our paper can then be solved for x_1 and the volume fraction ϕ_2 of the photosegregated phase. We need to assume here that the carrier diffusion lengths are larger than the feature sizes of the segregated morphology, so that we can assume constant photocarrier densities in the two phases. Taking $x_2 = 0.2$, room temperature, and fixing G by taking 1 sun illumination intensity we find $x_1 = 0.506$ and $\phi_2 = 0.020$, which is clearly smaller than the range $0.1-0.2$ mentioned by the reviewer. However, we find that ϕ_2 sensitively depends on x_2 . Taking $x_2 = 0.26$, we find $x_1 = 0.527$ and $\phi_2 = 0.103$, which is in this range. We think this shows that our theory should also be a good starting point for determining the final phase compositions and fractions. Because of the premature nature of these calculations we prefer to not include them in the present paper. In follow-up work we intend to address this issue in more depth.

7. Figures 3a-e presents phase diagrams for all studied cation compositions. While all phase diagrams qualitatively resemble each other, the FACs phase diagram is surprisingly different. This should be discussed more explicitly.

Indeed, the phase diagram of $\text{FA}_{7/8}\text{Cs}_{1/8}\text{Pb}(\text{I}_{1-x}\text{Br}_x)_3$ is very different from the others. There is even no critical point. Subtleties in the shape of the free energy curve for this compound in Fig. 1l, which have no simple explanation, are the reason for this. We have made our discussion about this at end of the first paragraph of the section "Phase diagrams" more explicit:

"The compound $\text{FA}_{7/8}\text{Cs}_{1/8}\text{Pb}(\text{I}_{1-x}\text{Br}_x)_3$ is special in the sense that it is stable in the dark for all values of x and T . Like $\text{MA}_{7/8}\text{Cs}_{1/8}\text{Pb}(\text{I}_{1-x}\text{Br}_x)_3$, the free energy curves are strongly asymmetric, but in contrast to $\text{MA}_{7/8}\text{Cs}_{1/8}\text{Pb}(\text{I}_{1-x}\text{Br}_x)_3$ no points of common tangent or inflection points occur,

which would be the locations of the binodals and spinodals, respectively; see Figs. 1l and o. We checked that this situation does not change for $T < 150$ K, which is the lowest temperature in Fig. 1, so that there is no critical temperature.”

8. The reviewer would like the authors to more explicitly clarify the discussion in Figure 3 regarding stable and unstable compositions and the effects of photosegregation. An examination and interpretation of the discussion regarding the solid green lines in Figure 3 would suggest the following: (a) photosegregation leads to universal terminal x values of $x \sim 0$ for all unstable stoichiometries. These are indicated by the dashed vertical green lines, (b) stable stoichiometries are temperature dependent. So whereas MAPb(I_{1-x}Br_x)₃ $x \sim 0.57$ is stable at 10⁻⁹f.u. at 300 K, it becomes unstable at $T < 300$ K under the same conditions. If correct, this should be stated more clearly as these are experimental testables of the model. Furthermore, the authors should comment on why a terminal $x \sim 0$ value at room temperature appears to differ from experimental terminal x values of $x \sim 0.2$ in MAPb(I_{1-x}Br_x)₃ (e.g. see Hoke et al. Chem. Sci., 2015, 6, 613) and analogous non-zero terminal x values in other systems. If kinetic constraints to reaching terminal x values of $x \sim 0$ are implicitly assumed then this should be stated in the text.

These considerations of the reviewer are completely correct. We stress again that our theory is thermodynamic in nature and considers the onset of phase segregation. The theory indeed predicts initial nuclei with $x \sim 0$. We are aware that experimental terminal values $x \sim 0.2$ are found and indeed attribute these to kinetic effects. In the paper of Ruth et al., already discussed at point 2 above, it is shown in kinetic Monte Carlo simulations of the phase segregation process that bromine atoms get kinetically trapped in the iodine-rich growing nuclei with a concentration $x \sim 0.2$. Initial nucleation with $x \sim 0$ is not incompatible with this observation. In fact, the photocarrier density threshold in the test calculations under point 6 above are performed for $x \sim 0$ at the onset of nucleation, see Eq. (4) of our paper. If we would take $x \sim 0.2$ in Eq. (4), a photocarrier density threshold is found that is about a factor 2,000 higher than the threshold measured by Elmelund et al. (Ref. 25), leading to nonsensical results in the above test calculations. We think that a value $x \sim 0$ at the onset of nucleation is perfectly compatible with a value $x \sim 0.2$ when phase segregation is complete. There is recent experimental evidence from a blue shift in photoluminescence measurements that phase segregation starts with a very pure I-rich phase, which gradually becomes less I-rich (Suchan et al, J. Luminescence 221, 117073 (2020), Babbe et al., J. Phys. Chem. C 124, 24608 (2020)). This is fully in agreement with this idea.

We added to the end of the section “Phase diagrams” the following paragraph:

“Our finding that $x \approx 0$ for the photosegregated I-rich phase (see the dashed green lines in Fig. 3) seems at odds with the experimental finding of Hoke et al. that $x \approx 0.2$ when segregation is complete [22]. An explanation for the latter finding was given by Ruth et al. [45]. In their kinetic Monte Carlo simulations of vacancy-mediated hopping of I and Br ions during the phase segregation process, Br ions get kinetically trapped in the I-rich nuclei, with a final concentration close to 0.2. Our theory applies to the onset of phase segregation and is therefore not incompatible with this result. There is recent experimental evidence from photoluminescence measurements that halide segregation commences with an almost I-pure phase, which gradually becomes less pure by inclusion of Br [47,48] (Suchan et al. and Babbe et al.). This is in line with our argument.”

We further clarified the discussion of these phase diagrams and of the experimentally testable aspects of the light-induced binodals by adding to the third paragraph of this subsection the sentence

“These binodals can be crossed by increasing the illumination intensity, but also by decreasing the temperature. The latter is a prediction of our theory that is experimentally testable.”

9. The authors predict the existence of triple points where 3 different phases would coexist. This prediction distinguishes this theory from other theories of photosegregation. Verifying the existence of these triple points through experiment would improve our understanding of photosegregation. However, these triple points are all predicted to be below room temperature where phase transitions may invalidate some assumptions of the model. The authors should therefore explicitly state which compositions are most-likely to have triple point that can be experimentally observed.

We think that $\text{MAPb}(\text{I}_{1-x}\text{Br}_x)_3$ is a good candidate to search experimentally for triple points. We added to the one-but-last paragraph of the section “Phase diagrams” the following sentences:

“ $\text{MAPb}(\text{I}_{1-x}\text{Br}_x)_3$ could be a good candidate to experimentally investigate the occurrence of triple points. It is predicted to have the highest critical temperature (266 K) of the investigated compounds. This has the advantage that the thermally activated motion of the halide ions is the least suppressed around the critical point, which facilitates the observation of the segregation. Down to 235 K, for which MAPbBr_3 shows a cubic to tetragonal transition [45] (Keshavarz et al.), no interfering structural transitions are expected. One can take x slightly higher or slightly lower than the critical Br concentration $x_c = 0.35$ to investigate the triple points of the type shown in Figs. 3f and p, respectively. By tuning the temperature and the illumination level these triple points can then be searched for by looking at, e.g., different features in the absorption spectrum.”

10. Following the authors’ conclusions about band gap differences in mixed-halide alloys dictating the phase stability, the reviewer concludes that alloys having small band gap slopes in Vegard’s law would be predicted to be more stable. i.e. $\text{CsPbI}_{1-x}\text{Br}_x$ should be the most stable out of all studied compositions. Following this line of thought, the reviewer would like the authors to simulate higher Cs loadings (e.g. 25%, 50%, 75%) into MA or FA based mixed halide perovskites to support their conclusions.

We followed this suggestion and performed additional calculations for $\text{MA}_{3/4}\text{Cs}_{1/4}\text{Pb}(\text{I}_{1-x}\text{Br}_x)_3$ (25% Cs). The results are given in Fig. S2 of the new Supplementary Note S4 (displayed below). It is observed that the trend when going from $\text{MAPb}(\text{I}_{1-x}\text{Br}_x)_3$ to $\text{MA}_{7/8}\text{Cs}_{1/8}\text{Pb}(\text{I}_{1-x}\text{Br}_x)_3$ is pursued. The critical temperature decreases further from 266 K ($\text{MAPb}(\text{I}_{1-x}\text{Br}_x)_3$, 0% Cs loading) and 244 K ($\text{MA}_{7/8}\text{Cs}_{1/8}\text{Pb}(\text{I}_{1-x}\text{Br}_x)_3$, 12.5% loading) to 216 K ($\text{MA}_{3/4}\text{Cs}_{1/4}\text{Pb}(\text{I}_{1-x}\text{Br}_x)_3$, 25% loading). The photocarrier density threshold for halide segregation is found to increase consistently with increasing Cs loading 0%-12.5%-25%. Both trends indicate a gradually increasing (photo)stability with increasing Cs loading. We note that the critical temperature for $\text{CsPb}(\text{I}_{1-x}\text{Br}_x)_3$ (250 K) is higher than that of $\text{MA}_{7/8}\text{Cs}_{1/8}\text{Pb}(\text{I}_{1-x}\text{Br}_x)_3$ and $\text{MA}_{3/4}\text{Cs}_{1/4}\text{Pb}(\text{I}_{1-x}\text{Br}_x)_3$, so that the trend regarding the critical temperature should reverse when the Cs loading is further increased. We performed these calculations, which are computationally intensive, only up to 25% Cs loading. For FA we expect similar results.

We added to the fourth paragraph of the section “Threshold photocarrier densities” the following:

“To investigate if the trend of increasing stability with Cs loading pursues, we show in Figs. S2d-f in Supplementary Note S4 phase diagrams in the dark and for photocarrier densities $n = 10^{-9}$ and 10^{-7} /f.u. of $\text{MA}_{3/4}\text{Cs}_{1/4}\text{Pb}(\text{I}_{1-x}\text{Br}_x)_3$, where the Cs loading is 25% instead of the 12.5% loading in $\text{MA}_{7/8}\text{Cs}_{1/8}\text{Pb}(\text{I}_{1-x}\text{Br}_x)_3$. The phase diagrams are very similar to the latter compound, but are shifted down in temperature. The critical temperature decreases from 266 K without Cs loading ($\text{MAPb}(\text{I}_{1-x}\text{Br}_x)_3$), to 244 K for 12.5% loading, and 216 K for 25% loading, indeed

showing a trend of increasing stability. This trend should be broken when increasing the Cs loading further, because for 100% Cs loading ($\text{CsPb}(I_{1-x}\text{Br}_x)_3$) the critical temperature is 250 K (see Fig. 3c). Figures S2g-i in Supplementary Note 4 show plots equivalent to Figs. 4a-c for $\text{MA}_{3/4}\text{Cs}_{1/4}\text{Pb}(I_{1-x}\text{Br}_x)_3$. The photocarrier density threshold is slightly higher than that of $\text{MA}_{7/8}\text{Cs}_{1/8}\text{Pb}(I_{1-x}\text{Br}_x)_3$, confirming the trend of increased photostability with increasing Cs loading.”

11. While the model findings suggest new insights into halide photosegregation, it would be best if these predictions were tested. The reviewer understands that the authors come from the modeling side. Consequently, it would seem that what could be done is for the authors to conduct an exhaustive literature search to find corroborating evidence for their model predictions.

During the writing of the manuscript we of course already conducted a literature search looking for corroborating evidence for our model predictions. We report about that in the fourth paragraph of the section “Threshold photocarrier densities”, starting with the sentence “Our results for the photostability of the different compounds agree with experimentally observed trends.” In a more exhaustive literature search we found the papers of Suchan et al. and Babbe et al., mentioned under point 5 above, which support our conclusion that halide segregation starts with an almost pure I phase, which gradually becomes less pure by inclusion of Br.

12. It would be useful if the authors ended their study by explicitly suggesting what optimal (stable) compositions for each system are.

We agree that is useful and added as last paragraph of the section “Threshold photocarrier densities”

“The consequence of the mechanism for light-induced halide segregation studied here is that the attractive band gap tunability of mixed halide perovskites at the same time leads to photostability problems. Nevertheless, routes towards optimal solutions follow from our study. For example, Fig. 4a shows that at 1 sun illumination and room temperature $\text{CsPb}(I_{1-x}\text{Br}_x)_3$ should be photostable up to 42% Br concentration. This allows, according to Fig. 2b, reaching a band gap of 1.94 eV. This is more than sufficient for the top layer in an efficient tandem solar cell, which has an optimal band gap of 0.96 eV for the bottom and 1.63 eV for the top cell. For $\text{MAPb}(I_{1-x}\text{Br}_x)_3$ and $\text{MA}_{7/8}\text{Cs}_{1/8}\text{Pb}(I_{1-x}\text{Br}_x)_3$, Br concentrations of about 33% and 35% can be reached, allowing band gaps of 1.73 and 1.78 eV, respectively, which are both still sufficient.”

Reviewer #2 (Remarks to the Author):

This topic has attracted a lot of attention due to the need for stable mixed halide perovskites. The paper presents a parameter-free theory showing how A-site alloying in mixed halide perovskites influences the thermodynamic stability under illumination. Construction of the phase diagrams for each compound in the dark and under illumination, distinguishing stable, metastable, and unstable regions is clearly described. The theory can be refined to include effects such as different recombination rates in the different phases and changes in the band gap due to strain, giving the paper additional impact. The authors have shown how they have gone beyond the large number of papers already published on this topic. I therefore recommend publication in Nature Communications.

There is a clarification that would help the reader understand the model which I would like the authors to address. At the bottom of page 5, it states 'The key ingredient of our unified theory for light-induced halide segregation is the consideration of the combination of the compositional free energy in the dark and the free energy of photocarriers in the presence of illumination' This sentence needs relating to the description on page 6 in the text and Figure 2a. Here, the authors argue that there is inward diffusion of I and outward diffusion of Br, which needs more information. I diffusion is driven by a lower concentration in the region towards which the I ions are diffusing, yet this region is I rich. Do we have to infer that an I ion that happens to reach an I rich region is captured due the region having a lower bandgap? How is this effect linked to the Br fraction variation of the bandgap presented in Figure 2b? What is the effect on the free charge carriers? Why should there be a net outward diffusion of Br? What is the effect of the illumination in addition to generating free carriers? The extent that these questions are addressed in the following text should be discussed at this stage.

We are happy that the reviewer recommends publication in Nature Communications and confident that we can answer the reviewer's questions.

The key idea of the mechanism driving the light-induced halide segregation is the following. In the absence of illumination there is a compositional free energy that plays exactly the same role as in usual phase separation processes of binary systems. In the dark, the system will try to minimize its free energy by motion of the halide anions. This leads to the phase diagrams in the dark displayed in Figs. 3a-e. Under illumination, photocarriers will be generated in the material. These photocarriers can minimize their free energy by moving to the phase with the lower band gap in the case of phase separation. We should now consider the total system of the perovskite with its distribution of halide anions and the generated photocarriers. Like any system, this total system will try to minimize its total free energy by the combined motion of

halide anions and photocarriers. This total free energy is given by Eq. (2) in the paper. The resulting phase diagrams are displayed in Figs. 3f-t.

The answers to the questions of the reviewer are the following:

Do we have to infer that an I ion that happens to reach an I rich region is captured due the region having a lower bandgap?

Answer: Yes, if this lowers the total free energy, which happens in the grey and pink regions of the phase diagrams. The reason is that by the growth of an I-rich region the photocarriers can lower their free energy because this region can now host more photocarriers.

How is this effect linked to the Br fraction variation of the bandgap presented in Figure 2b?

Answer: The Br fraction determines the band gap and thus the free energy of the photocarriers. The lower the Br fraction, the lower the photocarrier free energy.

What is the effect on the free charge carriers?

Answer: The free charge carriers, or photocarriers, have the tendency to move to the low-band gap phase in the case of phase separation and can then lower their free energy.

Why should there be a net outward diffusion of Br?

Answer: Because of the stoichiometry, the motion of one type of halide ions in a certain direction should be compensated by the motion of the other type in the opposite direction.

What is the effect of the illumination in addition to generating free carriers?

Answer: In our theory, this is the only effect of the illumination, and sufficient to explain the light-induced halide segregation.

These issues might indeed not have been clearly enough explained in the text. Particularly, what might have been insufficiently clarified is that the system will try to minimize the total free energy of the perovskite with its distribution of halide anions and the distribution of photocarriers by combined motion of halide anions and photocarriers. For this reason, we added to the beginning of the third paragraph of the section "Light-induced halide segregation" the following:

"The total system consisting of the perovskite with its distribution of halide anions and the generated photocarriers will try to lower its free energy by combined motion of halide anions and photocarriers. The free energy of the total system is the compositional free energy of the perovskite for a certain distribution of the halide anions (the free energy in the dark) and the free energy of a certain distribution of the photocarriers. ..."

To clarify the issue of outward motion of Br, we added to the first paragraph of this section:

"Because the stoichiometry cannot change, the inward diffusion of I should be accompanied by outward diffusion of Br."

Reviewer #3 (Remarks to the Author):

Chen et al present a thermodynamic model to describe photoinduced halide segregation (PHS) in perovskite films with different compositions ('A'-cations, halides). The critical photon flux threshold for phase segregation to occur is linked to the bandgap difference between the segregated phases which form under illumination.

The manuscript is well written and addresses a field of general interest and a phenomenon which currently limits the full exploitation of the wavelength tunability of lead halide perovskites. I recommend that this work could be published in Nature Comm after addressing the following points:

We are pleased that the reviewer thinks that our work could be published in Nature Communications and think that we can address the points mentioned by the reviewer.

1) Given that there are already a number of theories and models to explain PHS it would be great if the authors could point out more clearly what component of their theory has been described already and where the paper is breaking new ground.

In our general reply to the report of Reviewer #1 we mention that the idea that PHS (photo-induced halide segregation) is caused by funneling of the photocarriers to the low-band gap I-rich phase, which then drives further halide segregation, was put forward earlier by Draguta et al. in Ref. 23. A model for this was developed by Ruth et al. in Ref. 46. In our reply to point 2 of Reviewer #1 we mention that the model of Ruth et al. contains the carrier diffusion lengths of electrons and holes and a “geometrical band gap volume” as parameters, quantities that do not appear in our theory. We now address this in the fifth paragraph of the section “Threshold photocarrier densities”. Another important difference is that our theory provides a unified description of halide segregation in the dark and under illumination, leading to the complete phase diagrams in Fig. 3 and the prediction of novel phenomena such as the occurrence of triple points. Regarding the triple points, we added to the one-but-last paragraph of the section “Phase diagrams” ideas about how these could be experimentally verified. We think that with the generality of our theory and these novel predictions we are indeed breaking new ground. We think that these groundbreaking components are now sufficiently described in the revised manuscript.

2) What drives halide segregation in the first place when a perovskite crystal with homogeneously distributed halides is illuminated?

The halide segregation in a perovskite crystal with homogeneously distributed halides occurs in the same way as the segregation of the two components of a homogeneous binary mixture in an “ordinary” demixing process (for example oil-water demixing). In both cases there will be stochastic composition fluctuations that will grow to macroscopic regions of different composition when this leads to a lowering of the free energy. In our case, we should consider the total system of the perovskite with its distribution of halide anions and the photocarriers with their distribution. In the presence of illumination above a certain threshold, there will be a free energy lowering, because photocarriers can move to the low-band gap phase. This is the driving force for the halide segregation, which occurs in combination with a redistribution of the photocarriers. We think this is now well explained in the first three paragraphs of the section “Light-induced halide segregation” with the additions to the text made in response to the comments of Reviewer #2.

3) A ‘unified theory’ for PHS should also be able to explain the recent observation that halide segregation can be reversed at higher photon fluxes (Mao et al - Light-induced reversal of ion segregation in mixed-halide perovskites). How can this be achieved?

The illumination intensity at which the halide remixing takes place in that work (Ref. 37 in our manuscript) is about 2,000 sun, which corresponds to a photocarrier density of about $10^{-3}/f.u$ in the parent compound, $MAPb(I_{0.2}Br_{0.8})_3$, this leads, using Eq. (1) of our paper with $x_1 = 0.8$ and $x_2 = 0$ and the band gap relation for $MAPb(I_xBr_{1-x})_3$ displayed in Fig. 2b, to a photocarrier density of about $1.2 \times 10^6/f.u$. Obviously, this is far outside the applicability range of our theory. For example, we assume a Boltzmann distribution for the photocarriers,

which is only valid for photocarrier density lower than about $1/f.u.$ The explanation we have for the remixing at such high photocarrier densities is that the photocarriers will spill out to such an extent from a possibly nucleated phase into the parent phase that the driving force for halide segregation disappears. Without such driving force, we are essentially back to the situation in the dark, which favours mixing.

We added as sixth paragraph to the section “Threshold photocarrier densities” the following:

“It was recently reported that $MAPb(I_{0.2}Br_{0.8})_3$ remixes for an illumination intensity of 200 W cm^{-2} [37] (Mao et al.). This is equivalent to about 2,000 sun, which corresponds to a photocarrier density in the parent phase of about $10^{-3}/f.u.$ Using Eq. (1) and the band gap difference between $x_1 = 0.8$ (parent phase) and $x_2 = 0$ (nucleated phase), we would find at room temperature a photocarrier density in a potentially nucleated phases of about $1.2 \times 10^6/f.u.$ Obviously, our theory can no longer be applied at such extremely high densities, causing, among other things, a breakdown of the Boltzmann approximation used in Eq. (1). The explanation for the remixing could be that at these extreme densities there will be such a large spillover of photocarriers from the nucleated phase into the parent phase that the driving force for halide segregation disappears. This will essentially restore the conditions in the dark, where the mixed situation is favoured.”

4) Rational for the effect of Cs in PHS: The authors should summarise the concepts introduced in reference 33 more clearly to explain the effect of Cs.

We now summarize the concepts introduced in this reference by the addition of the following sentence to the second paragraph of the paper:

“In Ref. [33] (Bischak et al.) the reduced tendency for halide segregation when replacing MA by Cs was attributed to the smaller polarizability of Cs^+ as compared to MA^+ , which would reduce electron-phonon coupling and suppress halide segregation.”

We phrased our summary in this way to indicate that we do not necessarily agree with the explanation given in that work.

5) Page 11: the authors argue that ‘The light-induced nucleated phase is almost 100% I-rich’. This is very different from the experimental results in the following 2 references: Chem. Sci. 6, 613–617 (2015) and DOI 10.1038/s41563-020-00826-y, where the stabilised segregated I-rich phases are not I-pure phase. This would require some more discussion.

This is the same remark as made by Reviewer #1 at point 8. We think that the additions made to the manuscript in response to that remark also sufficiently address this request.

REVIEWERS' COMMENTS

Reviewer #1 (Remarks to the Author):

The reviewer has gone over the response of the authors and thanks the authors for addressing prior comments and concerns. The reviewer is satisfied with all of the changes made. The reviewer therefore recommends that the manuscript now be published in the Journal.

Reviewer #2 (Remarks to the Author):

The authors have answered in full the reviewers' concerns and made revisions to the paper that clarify the points raised. I recommend publication in Nature Communications without further revision.